# Does Maturity Change the Chemical-Bromatological Makeup of Cladodes in Spineless Forage Cactus?

Anandkumar Naorem [1], Mounir Louhaichi [2,3,*], Sawsan Hassan [4], Ashutosh Sarker [5], Shiva Kumar Udayana [6], Somasundaram Jayaraman [7] and Sachin Patel [1]

1   ICAR-Central Arid Zone Research Institute, RRS, Bhuj 370105, India
2   International Center for Agricultural Research in the Dry Areas (ICARDA), Tunis 1004, Tunisia
3   Department of Animal and Rangeland Science, Oregon State University, Corvallis, OR 97331, USA
4   International Center for Agricultural Research in the Dry Areas (ICARDA), Amman 11195, Jordan
5   Bangladesh Agricultural Research Council (BARC), Dhaka 1229, Bangladesh
6   College of Horticulture, Dr. YSR Horticultural University, Venkataramannagudem 534101, India
7   ICAR-Indian Institute of Soil Science, Nabibagh, Berasia Road, Bhopal 462038, India
*   Correspondence: m.louhaichi@cgiar.org

**Abstract:** In Kutch (Gujarat District, India), there is a growing concern about the lack of good quality forage owing to the arid climate and poor soil health. *Opuntia ficus-indica* has been increasingly recognized as a drought-resilient forage in arid Kutch. This study seeks to identify the maturity phase of cactus cladodes with the best forage qualities. Five accessions of spineless forage cactus (CBG, No. 1270, No. 1271, No. 1308, and Bianca Macomer) and three cladode maturity phases (young, intermediate, and mature) were examined in a randomized block design experiment in a 5 × 3 factorial arrangement. Although only mineral matter and total carbohydrate concentration were significantly different among the accessions, CBG showed better forage qualities than other accessions. Dry matter, organic matter, mineral matter, crude protein, ether extract, and total carbohydrate accumulations were higher in the intermediate phase. In the mature phase, relatively difficult to digest fiber components such as neutral detergent fiber, acid detergent fiber, lignin, cellulose, and hemicellulose increase. Our findings indicate that for spineless forage cactus grown in arid areas, the intermediate phase is the best phase to harvest cladodes for feeding livestock.

**Keywords:** CAZRI Botanical Garden; neutral detergent fiber; non-fiber carbohydrate; *Opuntia ficus-indica*; pectin





## 1. Introduction

The feasibility of annual crops is diminished due to irregular rainfall distribution and low precipitation, which also affects pasture and forage production and limits livestock production in arid regions [1,2]. One of the best choices for economic activity and livelihood improvement in arid parts of the world is livestock raising. However, livestock production methods in dry environments have lower forage availability and forage quality and providing only concentrated feeds to animals increases production costs [3]. In most cases, ruminant production systems in arid regions are an extractive activity, where productivity is the consequence of the intensive use of existing natural resources, causing the ecosystem to gradually deteriorate. With the introduction of species better suited to arid environments, there is a trend toward change in crop selection. Along with climate, it is well known that livestock pressure on already degraded rangelands accelerates the process of desertification. Parente and Parente [4] assert that the use of semi-extensive or extensive livestock in dry regions contributes to environmental changes because of the overcrowding of animals in areas beyond what the ecosystem can sustain.

Given the paucity of nutritious forage resources in Kutch (Gujarat District, India), one of the most significant issues restricting animal productivity is inadequate animal nutrition.

The competition between humans and animals for grain consumption is another constraining factor. This competition is intensifying as there is less pasture available than in the past. Similar cases have been reported by Filho et al. [2] in Nigeria and Njariu et al. [5] in Kenya. Thus, exploring the spineless forage cactus (*Opuntia ficus-indica*) can be a crucial strategic forage to use in production systems in arid regions of Kutch. In areas where the growth of other economic activities is constrained, spineless forage cactus is an excellent choice for supporting livestock [6]. Due to its photosynthetic mechanism known as crassulacean acid metabolism, cactus has shown excellent production in environments with little rainfall [7,8]. During the night, this photosynthetic pathway absorbs $CO_2$ and stores it as organic acid in vacuoles, where it is regenerated during the day to continue photosynthesis [9]. The stomata in the leaves of plants under drought stress close during the day to minimize evapotranspiration but open at night to absorb $CO_2$. As a result of this mechanism, spineless forage cactus use water more effectively than legumes and grasses [10,11]. Using spineless forage cactus has other benefits. For instance, cows who are given a diet that contains 50% spineless forage cactus and produce 15 kg of milk per day essentially do not need a drinking fountain [12]. Albuquerque et al. [13] also highlighted that inclusion of cactus pear silage up to 42% rate reduces water consumption in goats. The capacity of production systems would be increased by the use of spineless forage cactus in regions where it can grow normally and could serve as a main ruminant feed. This would reduce desertification, prevent the indiscriminate use of natural vegetation, and promote better adaptation to the harsh conditions of arid environments [14]. Livestock rumen can easily break down the cladodes of spineless forage cactus. This increases available energy, which promotes microbial development and digestion [15]. Using spineless forage cactus also decreases production costs and increases production efficiency [16]. In terms of ruminant feeding strategies, spineless forage cactus has positive outcomes, including a decrease in the quantity of concentrated feed required [17]. Wanderley et al. [18] examined diets for crossbred nursing cows that contained just 3.1% soybean meal, 34.2% roughage, and only 61% spineless forage cactus and found that they produced 11 kg of milk on average each day. In a different trial, Holstein heifers weighing an average of 243 kg were fed a daily base diet that included 1 kg of wheat bran and supplements of spineless fodder cactus (69.8%), sugarcane bagasse (27.6%), and urea (2.6%) and showed daily gains of 0.71 kg on average [19].

One of the main assets in any livestock operation is the availability of good quality forage. The nutritional aspects of forages affect animal production. Forage quality can differ not only between forage types but also within the same species or cultivars. Not every plant in a pasture has the same nutritive value, resulting in properties that can affect the chemical-bromatological composition of cactus cladodes indirectly or directly. The primary cause of declining forage quality is maturation [20]. As the plant matures and grows beyond its peak production phase, the production of fibrous components increases at the plant cell level, and this negatively affects the breakdown of forage in the rumen thereby affecting its digestibility. Understanding the chemical-bromatological makeup among spineless cactus accessions and maturation phases will help identify the best accessions with good forage quality and the best phase to harvest the cladodes. It will also aid in matching animal requirements and improving livestock performance economically.

The objectives of the current study are (a) to evaluate any differences in the chemical-bromatological composition of cactus cladodes among accessions, and (b) to understand whether maturity phases of cladodes change forage qualities.

## 2. Materials and Methods

Cladode samples were collected in 2021 at the research farm of ICAR-Central Arid Zone Research Institute (23°21′ N:69°77′ E, 15 asl), RRS-Bhuj, in the Kutch district of Gujarat, India. The experimental area is characterized as an arid zone owing to its low annual precipitation (mean annual precipitation = 424 mm in 2021) and high annual temperature (mean annual temperature = 26.4 °C, reached up to 45 °C during pre-monsoon

in 2021). The soil was characterized as sandy loam with the surface soil (0–30 cm) showing a soil pH of 8.45, electrical conductivity of 0.53 dSm$^{-1}$, exchangeable calcium content of 305 ± 12.44 milliequivalents per liter, sodium content of 55 ppm, and soil organic carbon of 0.19%. The experiment was laid out as a 5 × 3 factorial randomized block design (five accessions of forage and three phases of cladode maturity). Five of the best performing cactus pear accessions (*Opuntia ficus-indica*) were screened from 62 spineless forage cactus pear accessions available at ICAR-CAZRI, RRS-Bhuj (CBG-CAZRI Botanical Garden, accessions No. 1308, No. 1270, No. 1271, and Bianca Macomer). The spineless forage cactus accessions were planted in the month of July [21] in raised beds, with a spacing of 1 m (in the row) and 2 m (between the rows). Farmyard manure (0.7% N, 0.14% P, and 0.42% K) was applied to the soil at the rate of 5 t ha$^{-1}$ at an interval of 6 months. Weed management was carried out manually to avoid any contamination of chemicals in the fodder quality. Two-year-old plants of similar size were selected from each spineless forage cactus accession. The cladode samples were harvested at each maturity phase using the criteria detailed by Pessoa et al. [22]. The young cladodes of the plant are a bright green and develop on the sides or the ends of the plant. The intermediate cladodes, which are often located in the middle of the plant, are dark green in color. When completely developed, the cladodes take on a dark green hue with very slight yellowish undertones. Five plants of each accession were chosen based on the above-mentioned criteria to have their cladodes harvested. A single, representative cladode was collected from each plant at each maturity stage. This means that there were 25 samples collected at each stage of development (5 cladodes × 5 accessions). The cladodes were harvested with a sharp knife and the samples were washed with tap water followed by double washing with distilled water. The cladodes were chopped, oven-dried at 65 °C until a constant weight is achieved, weighed, passed through a stainless-steel grinder, sieved through 2 mm sieves, and stored in air-tight containers at room temperature, but were redried at 65 °C for 1 h and cooled before analysis.

There were two types of forage quality parameters: direct and derived chemical-bromatological composition. The dry matter DM (%) in the spineless forage cactus was determined using the oven drying method (105 °C until a constant weight was obtained) and was estimated by "100-weight loss on drying contents (%)" [23]. The organic matter (OM) and mineral matter (MM) were estimated by burning the cladode samples at 600 °C for 2 h [23]. MM represents the amount of inorganic residue after complete oxidation of the cladode sample. OM is the difference between DM and MM. Crude protein (CP) was analyzed using the Kjeldahl method [19], in which the sample was digested in $H_2SO_4$; followed by $NH_3$ distillation and titration of excess $H_2SO_4$. The ether extract (EE) was evaluated through Soxhlet extraction with petroleum ether [23]. Neutral detergent fiber (NDF) was determined by boiling the dried sample in a neutral detergent solution (Na-lauryl sulfate, EDTA, pH = 7.0) in a crucible and weighing the residue after a series of washing and drying [24]. Acid detergent fiber (ADF) was estimated by heating the dried samples in an acid detergent solution (cetyl trimethyl ammonium bromide in 1 N $H_2SO_4$) at room temperature followed by washing, drying, and weighing the residue [24]. Acid detergent lignin (ADL) is the residue obtained after treating the ADF fraction with $H_2SO_4$ (12 mol L$^{-1}$) [24]. A portable pH meter (Mettler Toledo, OH, USA) was used for pH analysis. The pectin content was isolated through a chemical extraction process by heating the dried samples in acidified water at 80 °C for 2 h [25]. Ash was estimated by using the dry oxidation method given by the Association of Official Agricultural Chemists [23].

The derived forage quality parameters are calculated as shown below:

(i) Total carbohydrate (TC) was calculated according to Equation (1) [26]:

$$TC = 100 - CP - EE - Ash \tag{1}$$

(ii) Hemicellulose (HEM) and cellulose (CL) were estimated according to the following Equations (2) and (3):

$$HEM = NDF - ADL \tag{2}$$

$$CL = ADF - ADL \qquad (3)$$

(iii) Total digestible nutrients (TDN) was calculated according to the methodology of Lofgreen and Meyer [27] (Equations (4) and (5)):

$$F = OM\,(100 + 0.000125EE) \qquad (4)$$

where F is a conversion factor

$$TDN = F * OM \qquad (5)$$

(iv) Non-fiber carbohydrates (NFC) was estimated according to Hall [28] as shown in Equation (6):

$$NFC = 100 - (CP + NDF + EE + Ash) \qquad (6)$$

All chemical-bromatological parameters are presented in percent unit of DM, except DM which was presented as percent of natural matter. All statistical analyses were performed using Rstudio [29]. Data normality was checked by a histogram, Q–Q plots, and a Shapiro–Wilk's test. The Shapiro–Wilk's test showed a $p$-value of less than 0.05 in all the parameters implying that the data were not normally distributed. Therefore, a Kruskal–Wallis test was run to explore the differences in chemical-bromatological parameters of five accessions of spineless forage cactus and three phases of cladode maturity differences. Spearman's rank correlation was used to measure the strength of association between the parameters. The trend values of each parameter between accessions and phases of cladode maturity were subjected to Dunn's multiple comparison tests. Significance was reported at the level of $p < 0.05$. A PCA biplot was generated by using the parameters to group the data into maturity phases of cladodes and spineless forage cactus accessions. Data visualization was performed using ggplot2 from R [30].

## 3. Results and Discussion

Changes in the chemical-bromatological composition of cactus cladodes and maturity phases were compared as shown in Tables 1 and 2. There was no distinct difference ($p > 0.05$) in dry matter (DM) content between cactus accessions and maturity phases of cladodes (Table 1). In our results, DM ranged between 12.84% and 14.29% of natural matter irrespective of accessions and cladodes maturity phase (Table 1). However, for two reasons, the majority of nutritionists favor using DM to determine the nutritional value of a feed or forage. First, the nutrients are used by the animal on a DM basis. Second, because all feeds can be compared on the same basis, it facilitates ration building. In contrast, low DM indicates a significant provision of water to animals through cactus [31] when water is a limiting resource in arid areas [32]. Likewise, differences in OM content among accessions and maturity phases of cladodes were comparable to DM.

Results indicated that there was a significant decrease ($p < 0.05$) in mineral matter (MM) with an increase in cladode maturity (Table 1). The Dunn's Multiple Comparison Tests revealed the maximum MM content during the young phase (median = 13.28%) followed by the intermediate (median = 12.35%) and gradually decreasing in the mature phase (median = 9.79%). The accessions showed notable differences in the case of MM content. The highest (median = 12.4%) and lowest (median = 7.06%) proportion of MM was presented by CBG and No. 1270, respectively. Santos et al. [33,34] found 12% and 11.9% MM in spineless forage cactus studies. An average cladode MM concentration (15.7%) was reported by Garcia et al. [35]. Spineless forage cactus frequently recorded high mineral concentrations attributed to the necessity for stomatal regulation and other physiological processes [36,37].

**Table 1.** Chemical composition of five forage spineless cactus accessions (CBG-CAZRI Botanical Garden, accessions No. 1308, No. 1270, No. 1271, and Bianca Macomer) at three different phases of cladodes maturity.

| Parameters (% Except pH) | Phase | Mean | SE | 95% Confidence Interval | | Median | Minimum | Maximum |
|---|---|---|---|---|---|---|---|---|
| | | | | Lower | Upper | | | |
| DM | Young | 13.72 | 0.08 | 13.56 | 13.87 | 13.77 | 13.02 | 14.29 |
| | Intermediate | 13.53 | 0.07 | 13.38 | 13.68 | 13.46 | 12.84 | 14.26 |
| | Mature | 13.55 | 0.08 | 13.40 | 13.70 | 13.61 | 12.86 | 14.14 |
| | Average | 13.60 | 0.04 | 13.51 | 13.69 | 13.64 | 12.84 | 14.29 |
| OM | Young | 66.33 | 0.59 | 65.18 | 67.48 | 66.79 | 61.72 | 70.20 |
| | Intermediate | 66.62 | 0.53 | 65.57 | 67.66 | 66.09 | 62.05 | 71.05 |
| | Mature | 66.82 | 0.55 | 65.73 | 67.90 | 66.83 | 62.39 | 71.04 |
| | Average | 66.59 | 0.32 | 65.96 | 67.21 | 66.76 | 61.72 | 71.05 |
| MM *# | Young | 12.54 | 0.45 | 11.66 | 13.41 | 13.28 | 8.19 | 14.59 |
| | Intermediate | 11.29 | 0.44 | 10.43 | 12.16 | 12.35 | 6.79 | 12.62 |
| | Mature | 9.79 | 0.40 | 9.00 | 10.58 | 9.79 | 6.32 | 12.54 |
| | Average | 11.21 | 0.28 | 10.66 | 11.75 | 12.14 | 6.32 | 14.59 |
| CP | Young | 4.05 | 0.09 | 3.87 | 4.22 | 4.08 | 3.34 | 5.11 |
| | Intermediate | 4.20 | 0.10 | 4.01 | 4.40 | 4.09 | 3.43 | 5.11 |
| | Mature | 4.25 | 0.09 | 4.07 | 4.43 | 4.21 | 3.36 | 5.10 |
| | Average | 4.17 | 0.05 | 4.06 | 4.27 | 4.11 | 3.34 | 5.11 |
| EE | Young | 1.32 | 0.01 | 1.29 | 1.34 | 1.32 | 1.23 | 1.43 |
| | Intermediate | 1.30 | 0.02 | 1.27 | 1.33 | 1.27 | 1.21 | 1.43 |
| | Mature | 1.31 | 0.01 | 1.29 | 1.34 | 1.30 | 1.22 | 1.43 |
| | Average | 1.31 | 0.01 | 1.29 | 1.33 | 1.29 | 1.21 | 1.43 |
| NDF * | Young | 15.64 | 0.24 | 15.18 | 16.11 | 15.78 | 13.47 | 17.42 |
| | Intermediate | 24.54 | 0.44 | 23.68 | 25.40 | 24.65 | 20.92 | 27.90 |
| | Mature | 37.21 | 0.57 | 36.10 | 38.32 | 37.50 | 31.85 | 40.90 |
| | Average | 25.80 | 1.06 | 23.73 | 27.87 | 24.65 | 13.47 | 40.90 |
| ADF * | Young | 11.27 | 0.21 | 10.85 | 11.69 | 11.30 | 9.49 | 12.75 |
| | Intermediate | 17.37 | 0.61 | 16.16 | 18.57 | 17.04 | 11.53 | 21.49 |
| | Mature | 26.71 | 0.15 | 26.42 | 27.01 | 26.51 | 25.50 | 27.95 |
| | Average | 18.45 | 0.77 | 16.94 | 19.96 | 17.04 | 9.49 | 27.95 |
| ADL * | Young | 0.82 | 0.05 | 0.72 | 0.91 | 0.87 | 0.47 | 1.21 |
| | Intermediate | 2.35 | 0.08 | 2.19 | 2.51 | 2.48 | 1.79 | 2.95 |
| | Mature | 3.32 | 0.15 | 3.03 | 3.61 | 3.56 | 1.94 | 4.19 |
| | Average | 2.16 | 0.13 | 1.90 | 2.42 | 2.18 | 0.47 | 4.19 |
| pH * | Young | 0.47 | 0.00 | 0.47 | 0.48 | 0.47 | 0.45 | 0.50 |
| | Intermediate | 0.44 | 0.00 | 0.44 | 0.44 | 0.44 | 0.43 | 0.45 |
| | Mature | 0.42 | 0.00 | 0.41 | 0.42 | 0.42 | 0.40 | 0.43 |
| | Average | 2.16 | 0.13 | 1.90 | 2.42 | 2.18 | 0.47 | 4.19 |
| Pectin * | Young | 0.44 | 0.00 | 0.44 | 0.45 | 0.44 | 0.40 | 0.50 |
| | Intermediate | 11.60 | 0.07 | 11.48 | 11.73 | 11.68 | 11.08 | 12.05 |
| | Mature | 14.97 | 0.04 | 14.89 | 15.05 | 14.93 | 14.65 | 15.30 |
| | Average | 11.84 | 0.29 | 11.27 | 12.40 | 11.68 | 8.23 | 15.30 |

* indicates that there is a significant difference in the variable between different phases of cladodes maturity in cactus pear accessions ($p < 0.05$). # indicates that there is a significant difference in the variable between different forage spineless cactus accessions ($p < 0.05$). pH is unitless. Variables with no superscript show no significant differences either among forage spineless cactus accessions or cladodes maturity phases. SE = standard error of mean; DM = dry matter; OM = organic matter; MM = mineral matter; CP = crude protein; EE = ether extract; NDF = neutral detergent fiber; ADF = acid detergent fiber; ADL = acid digested lignin.

**Table 2.** Derived forage quality parameters of five forage spineless cactus accessions (CBG-CAZRI Botanical Garden, accessions No. 1308, No. 1270, No. 1271, and Bianca Macomer) at three different phases of cladodes maturity.

| Parameters (%) | Phase | Mean | SE | 95% Confidence Interval | | Median | Minimum | Maximum |
|---|---|---|---|---|---|---|---|---|
| | | | | Lower | Upper | | | |
| TC *# | Young | 71.69 | 1.10 | 69.53 | 73.85 | 69.86 | 63.85 | 82.08 |
| | Intermediate | 71.76 | 1.09 | 69.61 | 73.91 | 70.41 | 61.94 | 80.68 |
| | Mature | 66.49 | 2.30 | 61.97 | 71.01 | 70.18 | 50.36 | 82.48 |
| | Average | 69.98 | 0.95 | 68.1 | 71.86 | 70.36 | 50.36 | 82.5 |
| HEM * | Young | 4.38 | 0.35 | 3.68 | 5.07 | 4.33 | 0.96 | 7.93 |
| | Intermediate | 7.18 | 0.72 | 5.76 | 8.59 | 7.53 | 1.56 | 15.26 |
| | Mature | 10.5 | 0.58 | 9.36 | 11.64 | 10.66 | 5.16 | 14.87 |
| | Average | 7.35 | 0.43 | 6.49 | 8.21 | 7.15 | 0.96 | 15.3 |
| CL * | Young | 10.45 | 0.22 | 10 | 10.9 | 10.48 | 8.5 | 12.25 |
| | Intermediate | 15.02 | 0.60 | 13.82 | 16.21 | 14.9 | 9.66 | 19.46 |
| | Mature | 23.39 | 0.18 | 23.02 | 23.76 | 23.63 | 21.91 | 25.22 |
| | Average | 16.29 | 0.66 | 14.99 | 17.58 | 14.9 | 8.5 | 25.2 |
| TDN | Young | 44.81 | 0.79 | 43.26 | 46.36 | 45.36 | 38.71 | 50.06 |
| | Intermediate | 45.16 | 0.72 | 43.74 | 46.58 | 44.34 | 39.16 | 51.38 |
| | Mature | 45.45 | 0.74 | 43.98 | 46.92 | 45.39 | 39.55 | 51.31 |
| | Average | 45.14 | 0.43 | 44.29 | 45.99 | 45.26 | 38.71 | 51.4 |

* indicates that there is a significant difference in the variable between different maturity phases in cactus pear accessions ($p < 0.05$). # indicates that there is a significant difference in the variable between different forage spineless cactus accessions ($p < 0.05$). Variables with no superscript show no significant differences either among accessions or maturity phases. TC = total carbohydrates; HEM = hemicellulose; CL = cellulose; TDN = total digestible nutrients.

There was no significant difference ($p > 0.05$) in crude protein (CP) and ether extract (EE) among the accessions and maturity phases of cladodes (Table 1). Because CP levels are crucial for the growth of cattle, ranchers must take them into account when deciding between various forage classes [38]. Although the high protein content might seem encouraging, in this and other studies CP levels for Opuntia grown under field circumstances were much lower. According to Edvan et al. [11], spineless forage cactus has CP levels ranging from 5.9 to 9.2%. Mayer and Cushman [38] found CP levels in greenhouse and field cladodes to be 26.4% and 7.1%. Proteins support the microbe-mediated breakdown of forage in the animal rumen. CP also makes up 60–80% of the total plant nitrogen [20]. Analysis of CP content in forage indirectly measures nitrogen concentration, which is calculated by multiplying CP by 6.25 (assuming nitrogen comprises approximately 16% of plant leaf protein). The average CP content of grains is 8 to 14%, with maize frequently falling between 7 and 9% and wheat and oats ranging between 12 and 14%. High protein feeds such as cottonseed meal and soybean meal frequently have a CP content between 40 and 50%. Hays often vary from 4 to 20% [39]. Despite accessions and cladode maturity phases, CP in spineless forage cactus was extremely low in our results (<6%), which represents the lowest quality forage according to general forage quality standards for livestock diets [35]. However, low CP levels can be improved through nutrient management practices. In Argentina, spineless forage cactus received large doses of fertilizer which resulted in a four-fold increase in biomass and a doubling of the CP levels [40]. A diet should contain concentrations of around 6–7% for the formation and development of ruminal bacteria, which are responsible for degrading slow-digesting nutrients [41,42]. Because microbes use urea for the synthesis of microbial protein when readily fermentable energy is present in the rumen, low CP concentrations can be improved [18]. According to a previous study on the development of cladodes, only the youngest pads had CP contents exceeding 15.0% and as the cladodes grew, the protein level decreased to 9.0% [43].

Although there was no significant difference ($p > 0.05$) in neutral detergent fiber (NDF) content of cactus accessions, the cladodes showed a trend for increasing NDF accumu-

lation with an increase in cladode maturity (Table 1). Dunn's multiple comparison tests show the maximum NDF content during the mature phase (median = 37.50%) followed by the intermediate (median = 24.65%) and gradually decreasing in the young phase (median = 15.78%). The total fiber content of the feedstock is represented by NDF. Usually, this is performed to help forecast the animals' feed intake. Intake is predicted to decrease when NDF increases. Unlike ADF, it is unrelated to quality and digestibility. In most forages, values typically fall between 50 and 80% [35]. Compared to grasses, legumes often have lower NDF and ADF levels and as forage maturity rises so do these values. Higher NDF values indicate higher fiber content in the forage sample. Therefore, cladodes with lower NDF are appropriate for feeding livestock [20]. NDF levels in cactus cladodes were between that of grains (average NDF = 10%) and grass straw (average NDF = 80%) [20]. In this study, NDF ranged from 13.5% to 40.90%, which reflects that cactus has a good forage quality [39]. When ruminants are fed with cactus pear with no other forage source, the animals experience diarrhea and weight loss due to low levels of digestible NDF [16]. Therefore, to increase productive performance, several authors recommend including a fiber source from other forage in animal diets containing spineless forage cactus [44,45]. To understand whether the level of NDF of cactus pear affects digestibility in goats, Pinho et al. [3] reported the minimum NDF to increase animal performance is 10.9%. The maximum neutral detergent fiber (NDF) concentrations were found in the mature phase of the accessions. This may be because cladodes were more developed in this phase. This is so the plant's cellular wall components (cellulose and hemicellulose), which make up the NDF, may grow more as it matures. This happens at the cost of organic molecules, which take part in metabolic activities through the deposition of non-nitrogenous organic molecules (cellulose, hemicellulose, lignin), resulting in a decrease in the concentration of nitrogen compounds [46].

Mature cladodes showed higher acid detergent fiber (ADF) (median = 26.51%) and acid digested lignin (ADL) (median = 3.56%) than young and intermediate cladodes (Table 1). In this study, ADF cladode concentrations ranged from 9.49% to 27.95% in several cactus accessions which were comparable to those reported by Santos et al. [47]. The amount of cellulose and lignin in the plant is represented by the ADF value. Because lignin is thought to be indigestible by animals, the ADF value is crucial since it is an indication of the portion of the feedstock that cannot be digested. The feedstuff becomes more difficult to digest as the ADF value increases. Forages have a greater ADF than grains and mixed diets. It can range from 3.0% in grains to 50.0% in grass straw [20]. In this study, ADF values were below 31%, which reflects its prime quality standard according to the general forage quality standards for livestock diets [39]. Through the dilution of fermentable food components such as starch, the fiber has the effect of prolonging chewing time, boosting saliva production, and decreasing the generation of fermentation acids [48]. ADL is the lignin fraction of ADF. Lignin has a deleterious effect on the nutritional availability of plant fiber, which is why it is regarded as a low-quality component in forages. By serving as a physical barrier to microbial enzymes, lignin prevents the digestion of cell wall polysaccharides such as cellulose and hemicellulose [49]. A Kruskal–Wallis test showed that the cladodes pH was not significantly different between cactus pear accessions. The difference in pH value was significant ($p < 0.05$) with a higher pH in young cladodes than in mature ones.

Overall, cladode maturity significantly affected the pectin content (chi-square (2) = 65.78, $p < 0.05$) (Table 1), but no significant difference ($p = 0.99$) was found in pectin content between accessions. In terms of pectin concentrations, there were variations between accessions and maturity phases of cladodes. Pectin, along with other compounds such as cellulose and hemicellulose, forms the structural foundation of a plant's cell wall. However, due to its high solubility, pectin helps increase the digestibility of DM and NDF [50,51]. Feeds with a high pectin content have significant potential for use in ruminant diets because they have a high energy density and fermentation takes place without the production of lactic acid, which helps maintain a balanced ruminal environment [52]. The species

and stage of a plant's development affect the polysaccharide concentration of the pectic fraction [53]. Pectin cannot be broken down by mammalian enzymes and needs to be broken down by microbes in the gastrointestinal system.

In the case of total carbohydrate (TC), no significant difference ($p > 0.05$) was found between the cactus accessions in either the young or intermediate phases. However, at the cladodes maturity phase, all the spineless forage cactus accessions showed significant differences in TC ($p < 0.05$) (Figure 1). When compared to the findings from other studies, the TC values in our study ranged between 50.36% and 82.5%. After two years of planting *Opuntia ficus-indica*, Menezes et al. [54] recorded 76.1%. In the genus Opuntia spp. of cactus, Wanderley et al. [18] recorded TC levels of 84.1%. Following extensive analyses of the TC content in spineless forage cactus, Sá et al. [55] recommended spineless forage cactus as a superior source of energy because it is rich in NFC.

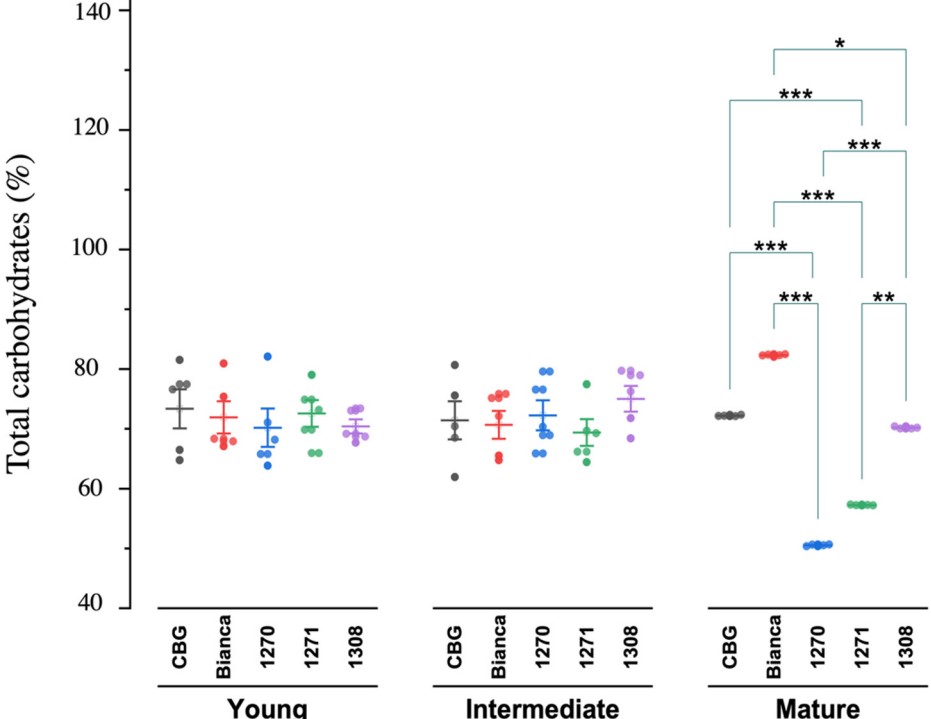

**Figure 1.** Comparison of total carbohydrate concentration (TC) in five forage spineless cactus accessions (CBG-CAZRI Botanical Garden, accessions No. 1308, No. 1270, No. 1271, and Bianca Macomer) at three different phases of cladodes maturity. The line over the data points indicates groups which were significantly different (Kruskal–Wallis test and Dunn's multiple comparison tests). The asterisk marks represent the significance level in each group (* at $p < 0.05$, ** at $p < 0.01$ and *** at $p < 0.001$).

Hemicellulose (HEM) and cellulose (CL) were calculated using some of the parameters discussed above. These derived parameters showed no significant differences ($p > 0.05$) among the accessions (Table 2). However, the maturity of the cladodes distinctly affected HEM and CL concentration ($p < 0.05$). HEM and CL significantly ($p < 0.05$) increased with the maturity of cladodes. In the case of HEM, the median value in the young phase was 4.33%, the intermediate phase 7.53%, and the mature phase 10.66%. In the case of CL, the median value in the young phase was 10.48%, the intermediate phase 14.9%, and the mature phase 23.63%. HEM is less digestible than CL. Our results demonstrated relatively higher CL than HEM content. The largest percentage of the cell wall component in most ruminant diets is cellulose, which is made entirely of -1,4-glucan [56]. Hemicellulose has a complex digestion process because it contains a variety of sugars and glucosidic links. Hemicellulose properties differ significantly between various plant cell wall types and

herbages [57]. When given a grass diet, ruminants digest more hemicellulose than cellulose, and when given a diet of legumes, they digest more cellulose than hemicellulose [58]. Hemicellulose's digestibility is negatively correlated with lignification and positively correlated with cellulose. The concentrations of hemicellulose and cellulose as well as how much they have been lignified are what primarily determine how digestible organic matter is [59].

No significant differences were observed in total digestible nutrients (TDN) values between accessions and maturity phases of the cladodes ($p > 0.05$) (Table 2). The values ranged between 38.71% and 51.4%. TDN is derived from the ADF value and indicates the overall digestibility of the forage. Total digestible nutrients (TDN) is a measure of the roughage or feed estimated energy content. It is determined using a formula that takes into account the feedstuff's ADF, NDF, and CP to provide an approximation of its energy content. Usually, the feedstuff is regarded as more energy-dense the higher the value. Higher grade hays typically range from 50% to 60% TDN, whereas lesser quality hays are often in the 40 to 50% range. Certain hays and legumes may occasionally contain 60 to 70% TDN. Grain and grain mixtures typically contain between 70 and 80% TDN [39].

Non-fiber carbohydrates (NFC) showed significant differences among the phases of cladodes maturity ($p < 0.05$) (Figure 2). NFC ranged between 49.3% and 80.1%, irrespective of accessions. The high NFC content of the spineless forage cactus is thought to cause high rumen degradability, which can result in over 80% of the DM vanishing within 48 h of incubation [2]. Non-structural carbohydrates (sugars and starch), neutral detergent soluble fiber (fructans, glucans, and pectin), and organic acids are known components of the NFC fraction that can affect the rumen fermentation pattern [28]. The high energetic value and high concentration of NFC in cactus were also reported by Bispo et al. [44] and Costa et al. [60]. Due to its high NFC concentration and considerable impact on feeding cost reduction, including high amounts of cactus pear in diets minimizes the need for energy concentrates.

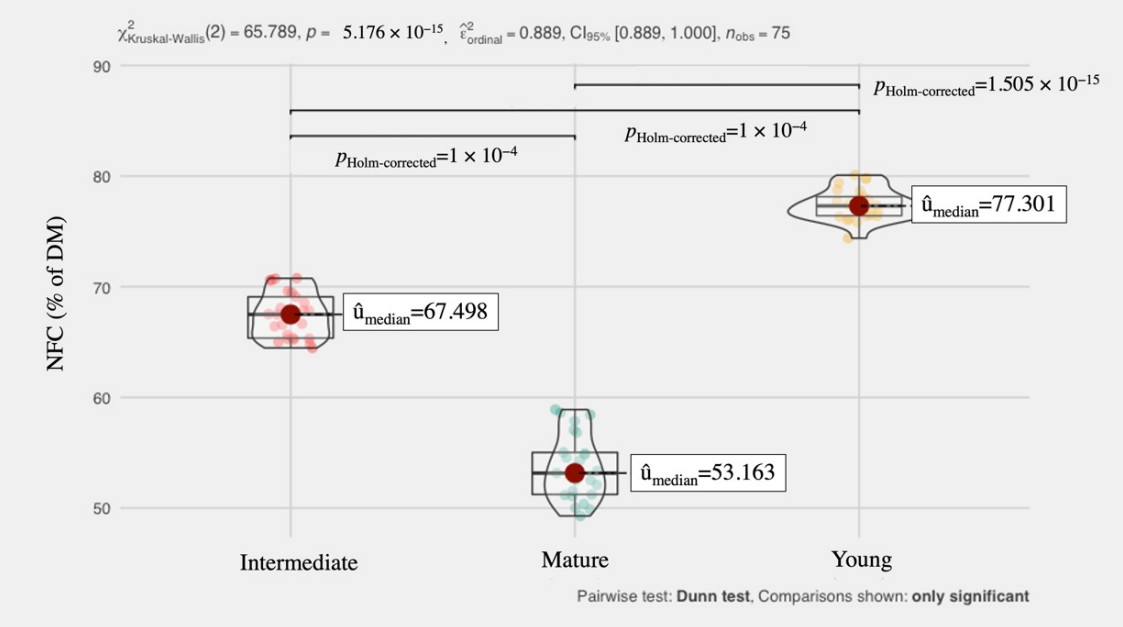

**Figure 2.** Non-fiber carbohydrate content (NFC) of five forage spineless cactus accessions (CBG-CAZRI Botanical Garden, accessions No. 1308, No. 1270, No. 1271, and Bianca Macomer) at three different phases of cladodes maturity. The lines over the violin plots represent the significant difference between the maturity phases ($p < 0.05$). Since the data is not normally distributed, Kruskal–Wallis test and Dunn's multiple comparison tests were used to analyze any differences in NFC among maturity phases; $n = 25$ (5 cladodes × 5 accessions) refers to the number of cladode samples from each accession at a specific maturity phase.

Correlation studies between the forage parameters revealed 44 significant correlated pairs (excluding r = 1) ($p < 0.05$) and 76 non-significant pairs ($p > 0.05$) (Figure 3). Since the data did not follow a normal distribution, a non-parametric Spearman's rank correlation test was computed to assess the relationship of forage quality parameters. A strong positive correlation was present between NDF and ADF (r = 0.87, $p < 0.001$). Pectin was positively correlated with NDF (r = 0.92, $p < 0.001$), ADF (r = 0.88, $p < 0.001$), ADL (r = 0.83, $p < 0.001$), and CL (r = 0.86, $p < 0.001$).

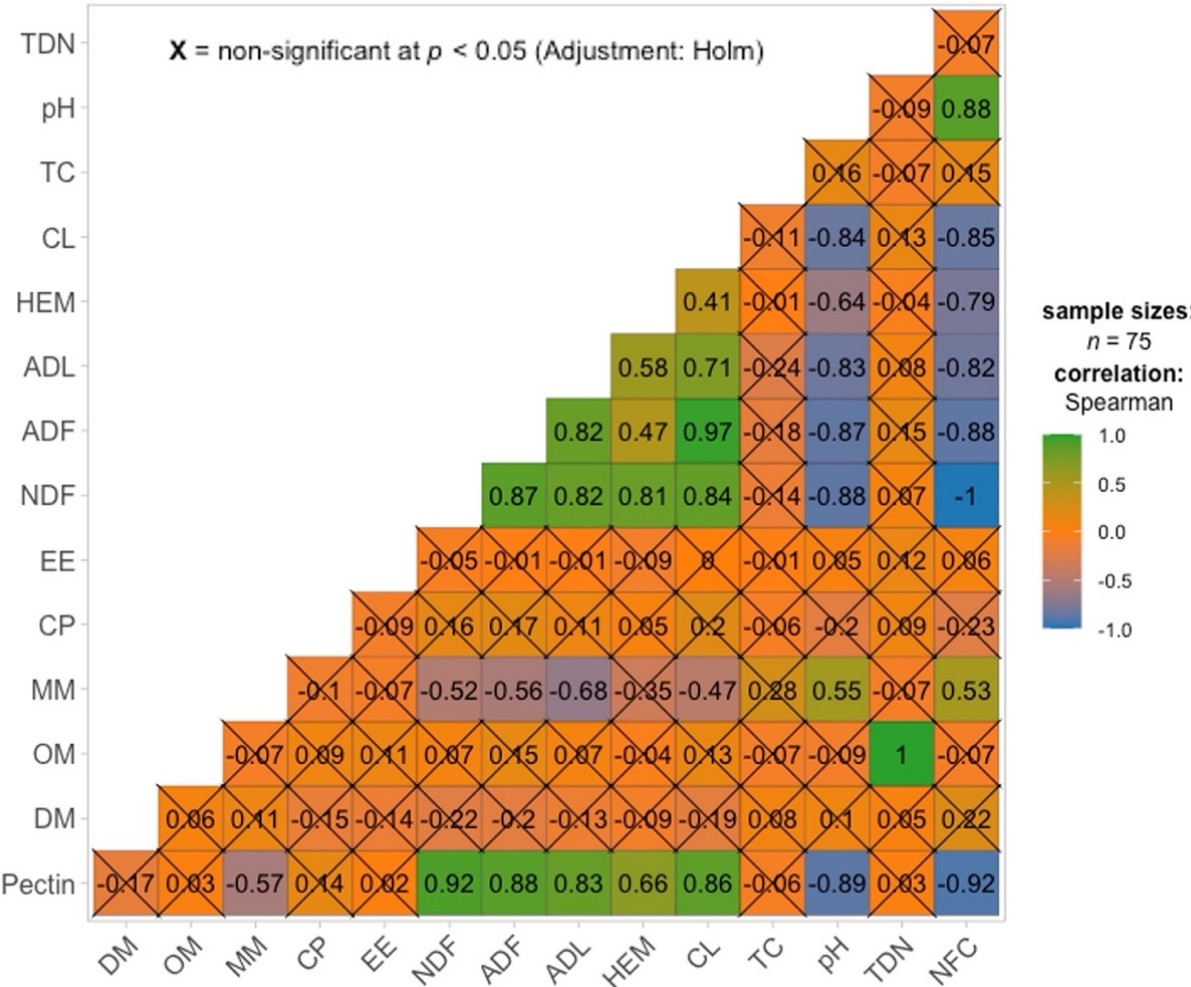

**Figure 3.** Correlogram depicting the correlation between forage quality parameters in five forage spineless cactus accessions (CBG-CAZRI Botanical Garden, accessions No. 1308, No. 1270, No. 1271, and Bianca Macomer). The color band represents the strength of the correlation. 'X' indicates a non-significant correlation ($p > 0.05$). DM = dry matter; OM = organic matter; MM = mineral matter; CP = crude protein; EE = ether extract; NDF = neutral detergent fiber; ADF = acid detergent fiber; ADL = acid digested lignin; HEM = hemicellulose; CL = cellulose; TC = total carbohydrates; TDN = total digestible nutrients; NFC = non-fiber carbohydrates.

The grouped PCA biplot shows the higher concentration of each parameter in different maturity phases of the cladodes. The pH was the only parameter found to be the highest during the young phase. The parameters (with high concentration = favorable) such as MM, TC, DM, EE, OM, and CP were concentrated within the intermediate group. Parameters with high concentration = not favorable, such as HEM, NDF, ADF, pectin, CL, and ADL, were concentrated within the mature phase (Figure 4). This indicates the intermediate phase is the best cladodes maturity level at which to harvest and feed livestock. In terms of accessions, CBG showed higher forage quality than other accessions.

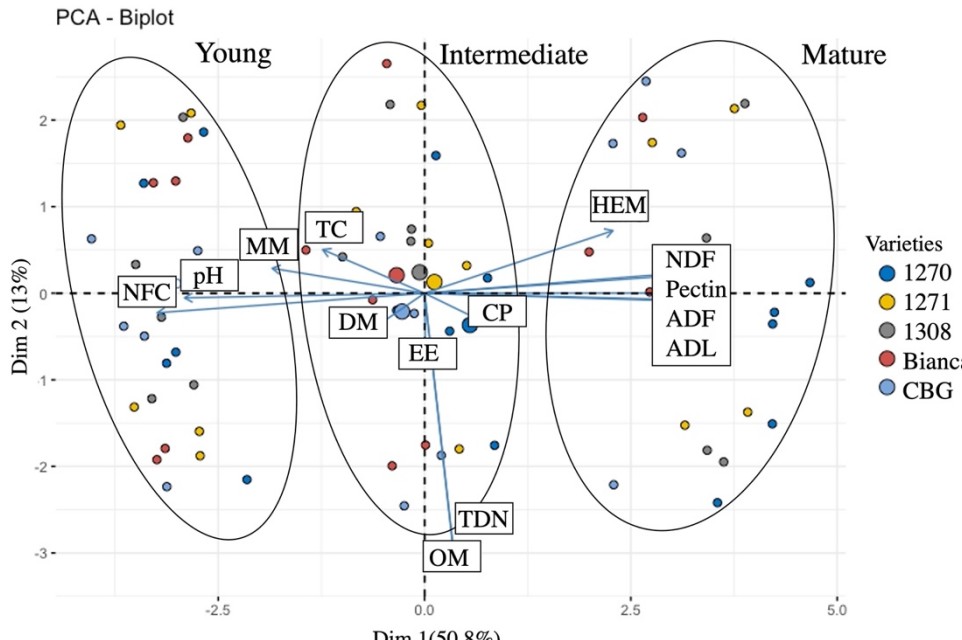

**Figure 4.** Principal component analysis-biplot (PCA-biplot) of the chemical-bromatological parameters of cactus pear accessions. The data points have been grouped by the maturity phase of the cladodes of five forage spineless cactus accessions (CBG-CAZRI Botanical Garden, accessions No. 1308, No. 1270, 1271, and Bianca Macomer). MM = mineral matter; NFC = non-fiber carbohydrates; TC = total carbohydrates; DM = dry matter; EE = ether extract CP = crude protein; TDN = total digestible nutrients; OM = organic matter; HEM = hemicellulose; NDF = neutral detergent fiber; ADF = acid digested lignin; ADL = acid detergent fiber.

## 4. Conclusions

This research extends our knowledge of cactus farming as a drought-resilient forage in arid soils. The findings suggest forage spineless cactus cladodes in the intermediate phase have the best forage quality. Regardless of the tested accessions, the results show the high nutritive value of the spineless forage cactus. It has low levels of dry matter, fiber, and protein but is a significant supply of non-fiber carbohydrates, pectin, minerals, and fresh water. As a result, it is used as an alternative feed for animals in arid areas. Based on these findings, spineless forage cactus should be used as the main source of nutrition for ruminants in arid areas. Since Kutch is an area known for its milk production, this research raises questions worth further investigation. For example, the effects of a cactus diet on milk quality and quantity should be assessed. Furthermore, it would be worthwhile to assess ecosystem services from cactus farming in arid soils owing to its relatively poor soil health.

**Author Contributions:** Conceptualization, A.N.; methodology, A.N. and S.J.; software, A.N., S.K.U., S.P. and S.J.; validation, S.H., M.L. and A.S.; formal analysis, A.N. and S.J.; investigation, A.N. and S.P.; resources, A.N., S.H. and M.L.; writing—original draft preparation, A.N.; writing—review and editing, S.H., M.L. and S.J.; visualization, A.N.; supervision, M.L., S.H., A.S., A.N.; project administration, A.S., M.L. and S.H.; funding acquisition, A.S. and M.L. All authors have read and agreed to the published version of the manuscript.

**Funding:** This research was funded by the Indian Council of Agricultural Research and the CGIAR Research agreement No. 200091 and Program on Livestock Agri-Food Systems agreement No 200173.

**Institutional Review Board Statement:** Not applicable.

**Informed Consent Statement:** Not applicable.

**Data Availability Statement:** Data sharing is not applicable to this article.

**Acknowledgments:** This study was conducted within the framework of the collaborative research program between the Indian Council of Agricultural Research (ICAR) and the International Center for Agricultural Research in the Dry Areas (ICARDA). Special thank goes to the Bhuj Central Arid Zone Research Institute (CAZRI) Regional Research Station for implementing the study and to the Livestock, Climate and System Resilience (LCSR) of the OneCGIAR for their support. The opinions expressed in this work belong to the authors and do not necessarily reflect those of ICAR, ICARDA or the OneCGIAR.

**Conflicts of Interest:** The authors declare no conflict of interest.

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
