# Peer review of "Does Maturity Change the Chemical-Bromatological Makeup of Cladodes in Spineless Forage Cactus?"

_sustainability, doi:10.3390/su141811411_

Round 1

Reviewer 1 Report

Dear Authors,

the manuscript reviewed holds merits. My only concernings are related to:

a) material and methods must be better described;

b) in such experimental design with no controled environment as light, temperature and so on, I would expect at least a new round of repetition from all experiment, or at least main analysis. I mean, could the same results repeted in different enviromental conditions?

Author Response

Dear Authors,

the manuscript reviewed holds merits. My only concernings are related to:

  1. material and methods must be better described;

Response: Noted and the materials and methods section are revised with detailed description for each of the fodder parameters. Specific details to rainfall and temperature in Kutch district were added. Other suggested corrections were also made.

  1. b) in such experimental design with no controled environment as light, temperature and so on, I would expect at least a new round of repetition from all experiment, or at least main analysis. I mean, could the same results repeted in different enviromental conditions?

Response: Repeating the whole experiment in a controlled environment such as incubation chamber or microcosm experiment will need a different setup and the whole study has to be revised. The main goal of the research was to understand which phenological stage is best in forage quality, when planted in arid soils under natural conditions of arid climate. The cladodes were planted in the month of July, which is the best time of planting cactus pear in Kutch region. Since Kutch is predominated by sandy loam soil with high pH, specific sites were chosen for this experiment to understand the behavior of the cactus pear in these conditions. We are planning to conduct cactus pear experiments under farmers field conditions in near future.

Reviewer 2 Report

The description of the methodology should be improved. Report the methods used. It needs to be corrected. Not readable figures. Especially No. 4. No comparison of the varieties.  A number of comments were given in the text of the article.

Author Response

Comment: The description of the methodology should be improved. Report the methods used. It needs to be corrected. Not readable figures. Especially No. 4. No comparison of the varieties.  A number of comments were given in the text of the article.

Response: The materials and methods section is revised with detailed description for each of the fodder parameters. Specific details to rainfall and temperature in Kutch district were added. Other suggested corrections were also made. Figure 4 is enhanced with the addition of colour grouping based on the accessions.

Line 97: The methodology must be completed.

Response: The materials and methods section is revised and missing information is added.

Line 102: What kind of unit is it?

Response: The unit is milliequivalents per litre; the unit is clarified in the revised manuscript.

Line 117: What temperature?

Response: The sampling procedures and temperature were added.

Line 120: What do these numbers mean?

Response: These numbers represent the method code number assigned by the Association of Official Analytical Chemists [19]. These numbers were  removed in the revised manuscript.

Line 127: You should write down what the methods of analysis are. What were the conditions for the extraction, what solutions.

Response: The analysis procedures were added.  

Line 150: The data visualization wa

Response:  The comment is not clear

Line 154: Transfer this sentence to the methodology. Up to line 104.

Response: This sentence is deleted from Results section and similar sentence is revised in Materials and Methods section.

Line 155: This sentence is not a discussion but a general idea. It is not a comparison of the results.

Response: This sentence is deleted.

Line 167: Explain the abbreviation ”MM”

Response: The abbreviation explanation (mineral matter) is added in the revised manuscript.

Line 171: Once written, you don't have to repeat it.     p<0.05

Response: “p<0.05” deleted

Line 250: Explain the abbreviation. “TC”

Response: The abbreviation explanation (Total carbohydrate) is added in the revised manuscript.

Reviewer 3 Report

Dear author

Please fin attached pdf file for comments. 

Regards

Author Response

Title: Title should be modified as to remove the word “alter”

Response: “alter” replaced by “change”

Line 29: “soils”

Response: “soils” replaced by “areas”

Line 59: “alter”

Response: “alter” replaced by “change”

Line 100: Give means of rainfall and temperature

Response: The information is added in the revised MS

Line 106: Give more details regarding these varieties/cultivars. Are these belonging to O. ficus indica ?

Response: All are Opuntia ficus-indica accessions 

Line 107: in addition to size, age is more meaningful than size

Response: all plants were 2 years old and planted at the same period,  The following sentences were added to clarify the section:

The cladodes were planted in raised beds, with a spacing of 1m (in the row) and 2m (between the rows). Farmyard manure (0.7% N, 0.14% P and 0.42 % K) was applied to the soil at the rate of 5 t ha-1 at an interval of 6 months. Weed management was carried out manually to avoid any contamination of chemicals in the fodder quality. Two-years old plants of similar size were selected from each forage cactus accession.

Line 107: How many cladode/maturity phase?

Response: Five plants of each accession were chosen based on the above-mentioned criteria to have their cladodes harvested. A single, representative cladode was collected from each plant at each maturity stage. This means that there were 25 samples collected at each stage of development (5 cladodes X 5 accessions)

Line 108:  It will be better if you specify these criteria even you cite ref.  Also avoid plagiarism with Brazilian study (Pessoa et al)

Response: The paragraph is rewritten and more details about the criteria of maturity phases were added.

Line 115: Describe how were your samples? are pieces of cladodes or entire cladodes ? Specify T° and duration of drying

Response: Details about sampling and samples handling were added

Line 130: standardize: Eq. or eqn.

Response: Corrected (Eq. abbreviation is used)

Line 140: how do get data related to ash?

Response: Information about Ash estimation is added: “Ash is estimated by using the dry oxidation method given by the Association of Official Agricultural Chemists [19]”.            

Line 156: To be moved at the end the results presentation” The chemical-bromatological compositions were notably close to results reported by Abidi et al. [27] and Siqueira et al. [28].”

Response: The sentence is removed, and the following citation numbers were changed accordingly.

Line 159: Table 1 did not show these values

Response: Table 1 is revised, and all missing values are added.

Line 169: what is the usefulness of table 1 if the values discussed within the text were nit showed?

Response: Table 1 is revised, and all missing values are added.

Line 174: Table 1 didn’t show the results discussed here

Response: This data is the part of the discussion, reported by Garcia et al. [33].

Line 181: “Opuntia” itaic

Response: “Opuntia” is italicized.

Line 183: 26.3% should be 26.4

Response: All values were checked and corrected.

Line 191: according to table 1: < 4.5%

Response: All values were checked and corrected.

Line 200: replace “forage” by “cladodoes”

Response:  This done as suggested.

Table 1: what des it mean: IQR”

Response: IQR refers to Interquartile range. In the revised table 1, the IQR is replaced by minimum and maximum values since these are explained in text.

Line 219: Impossible to follow! where the readers can pick this information in your manuscript ?

Response: Table 1 is revised, and all missing values are added.

Line 236: species or varieties?

Response: All planting materials are accessions, correction is made in the revised manuscript.

 Line 255: “Opuntia” itaic

Response: “Opuntia” is italicized.

Line 288, 289, and 301 check the highlighted numbers and confirm

Response: All numbers were checked and corrected.

Table 2: what does it mean: IQR”

Response: IQR refers to Interquartile range. In the revised table 2, the IQR is replaced by minimum and maximum values since these are explained in text.

Figure 2: what are n refer to?

Response: The description of n is added in the caption of Figure 2.

Line 331: Was that test mentionned in Mat&Met ?

Response: The Materials and Methods section was revised, and all missing information were added.

Line 332: 57% can not be considered as a solid correlation

 Response: the sentences is deleted.

Line 334: 87% not 86%

Response: All values were checked and corrected.

Line 336: check highlighted number

Response: All values were checked this value is deleted in the revised manuscript.

Line 341: Mention PCA in Mat&Met section

Response: The Materials and Methods section was revised, and all missing information were added.

Line 356: This is not an original finding

Response: this sentence is deleted in the revised manuscript.  

Line 363-364: Line out of the scope of this paper

Response: This sentence is deleted in the revised manuscript.

Round 2

Reviewer 3 Report

Dear Author

Still one one typing error in line 226 (34 instead of 340).

Regards